# Microplastics in Widely Used Polypropylene-Made Food Containers

**DOI:** 10.3390/toxics10120762

**Published:** 2022-12-07

**Authors:** Jun Hu, Xin Xu, Ying Song, Wenqi Liu, Jianqiang Zhu, Hangbiao Jin, Zhu Meng

**Affiliations:** 1College of Environment, Zhejiang University of Technology, Hangzhou 310032, China; 2Innovation Research Center of Advanced Environmental Technology, Eco-Industrial Innovation Institute ZJUT, Quzhou 324018, China; 3Quzhou Municipal Bureau of Ecology and Environment, Quzhou 324007, China; 4Hangzhou Branch, Shaoxing Industrial Science Design Research Institute Co., Ltd., Hangzhou 310052, China

**Keywords:** microplastics, takeout food containers, plastic container, polypropylene, human intake

## Abstract

As a potential threat to human health, ingestion of microplastics (MPs) has become of concern. Limited studies have carefully characterized the occurrence of MPs in polypropylene-made takeout food containers (TOFCs), which have been widely used in China. In this study, TOFC samples (*n* = 210) were collected from seven Chinese cities (Hangzhou, Guangzhou, Shanghai, Xining, Chengdu, Qingdao, and Dalian) and analyzed for MPs. All the TOFC samples contained MPs, with an abundance of 3–43 items/TOFC. The TOFCs from Chengdu (25 items/TOFC) contained the highest mean abundance of MPs, which is significantly (*p* < 0.01) higher than that from Shanghai (8.7 items/TOFC). Fiber accounted for 66–87% of the total for the shape of the MPs in the TOFCs from the different Chinese cities. Most of the MPs in the TOFCs from the different cities had a size of 201–500 μm and accounted for a mean 34–42% of the total MPs in the TOFCs. The major color type of the MPs in the TOFCs was transparent, accounting for a mean 39 (Qingdao)–73% (Hangzhou) of the total MPs. Polymer compositions of the MPs in the TOFCs were consistently dominated by polypropylene, which represented a mean 56–73% of the total MPs. The estimated daily intake of MPs for the general Chinese population through using TOFCs was in the range of 0.042–0.14 items/kg bw/day. To our knowledge, this is the most comprehensive study investigating the occurrence of MPs in TOFCs from China, which contributes to a better understanding of the sources of human oral exposure to MPs.

## 1. Introduction

Global production of plastics is still increasing, and had reached approximately 380 million tons in 2020 [1,2] Plastics have numerous applications in consumer products and industrial materials [3,4,5]. Degradation of these plastic products occurs in the environment and has generated numerous microplastics (MPs), which are defined as plastic particles with the diameter of 1 nm–5 mm (length of 3 nm–15 mm and length to diameter ratio of >3 for fibers) [6,7]. Currently, MPs are widely present in the global environment, including surface water, sediments, and the atmosphere [8,9,10,11,12], and humans are ubiquitously exposed to MPs through many pathways [13]. Evidence on laboratory animal experiments suggests that the intake of MPs could induce local immune responses, as well as energy and lipid metabolism disruption [14,15,16]. Moreover, MPs also serve as vectors that could transport toxic organic pollutants, bacteria, and even viruses around the global environment [17,18,19]. In recent years, MP pollution has emerged as a potential threat to human health [20,21].

Understanding the sources of human exposure to MPs is crucial to accurately evaluating the risk of human exposure to MPs [22]. Intentional or unintentional human intake of MPs occurs primarily through oral ingestion and inhalation, as proven by the detection of MPs in human stools, lungs, and indoor dust [23,24,25]. Recent monitoring studies have demonstrated the presence of MPs in various drinks and foods for human consumption, including table salts, teabags, drinking water, and sea fish [22,26,27]. Based on these studies, it has been estimated that a quantity of as high as 39,000–52,000 items of MPs could be ingested by humans every year from various food stuffs [13]. Food packaging has been considered an important source of MPs occurring in foods seeing as many food products, especially for takeout foods, are packaged with plastic materials [28,29].

In China, the fast-food industry has been increasing rapidly in order to meet the demands of modern life [30,31]. Disposable takeout food containers (TOFCs) made of polypropylene (PP) have been widely used to package fast food in Chinese restaurants because of PP’s convenience and low cost [32]. Moreover, limited studies have isolated and characterized MPs in PP-made food containers from China [33,34]. The COVID-19 pandemic is also contributing to the increased use of PP-made takeout food containers [35,36]. Polypropylene plastic is manufactured by polymerizing the propylene monomer [37]. This manufacturing process generates a great number of MPs, and the residual MPs in commercial TOFC products is transferred to warm foodstuffs [38]. This has been considered as a potential threat to human health.

To our knowledge, few studies have characterized the occurrence of MPs in various plastic food containers collected from China [33,34,39]. For instance, employing the spectroscopic technique (FT-IR), Fadare et al. (2020) reported that the average weight of isolated MPs was 3–38 mg/pack in consumer PP-made food containers (*n* = 150) from Beijing, China. Du et al. (2020) analyzed takeout food containers (*n* = 60) made of four common polymers from five Chinese cities, and reported that the abundance of MPs ranged from 3 to 29 items/container. However, these previous studies usually included only a relatively small sample size and did not analyze the characteristics (e.g., size, color, shape, and chemical composition) of the MPs. More studies with a larger sample size are still warranted to comprehensively understand the pollution of MPs in PP-made TOFCs from China. Such information is necessary for understanding the sources of human oral exposure to MPs, and for risk assessment on human exposure to MPs.

In this study, polypropylene-made TOFCs (*n* = 210) used for fast-food packaging were collected from restaurants in seven Chinese cities (Hangzhou, Guangzhou, Shanghai, Xining, Chengdu, Qingdao, and Dalian) with the aim of investigating the abundance, shape, size, color, and chemical composition of MPs in TOFCs. The human daily intake of MPs was also estimated for the general Chinese population, based on the abundance of the MPs in the TOFC samples.

## 2. Materials and Methods

### 2.1. Sample Collection

In total, 210 clean TOFC samples were collected from Chinese food restaurants in 7 Chinese cities including Hangzhou (*n* = 35), Guangzhou (*n* = 21), Shanghai (*n* = 30), Xining (*n* = 37), Chengdu (*n* = 27), Qingdao (*n* = 28), and Dalian (*n* = 32) between April and August, 2020. Dalian, Qingdao, Guangzhou, Hangzhou, and Shanghai represent typical populated coastal cities in China while Xining and Dalian represent typical inland Chinese cities. The restaurants were located in the downtown areas of these cities; detailed information on the sampling regions is shown in the Appendix A (Appendix A). The mean size of a typical TOFC is 27 × 11 × 5.7 cm. The TOFCs from these restaurants are mainly used to wrap up fast food, including cooked rice, meat, vegetables, noodles, and soups. Despite the recycling symbol showing that the TOFCs were produced using polypropylene, the chemical compositions of TOFCs were still further identified using Fourier Transformed Infrared (FT-IR) spectrometry (Nicolet™ iS 50; Thermo-Scientific, Waltham, MA, USA). Typical spectrums of the TOFCs consistently showed that they were made of polypropylene (Appendix A, Appendix A). Detailed information on the TOFC samples (e.g., identification code and recycling symbol) is shown in Appendix A (Appendix A). Based on the identification code information of the TOFCs, it is believed that the majority of TOFCs are made in different ways. In the TOFC sampling process, we avoided collecting TOFC samples produced by the same manufacturers.

The TOFCs were always packed, stacked together, and stored in the cupboards of the restaurants. In this study, in order to reduce the background contamination of the MPs from the atmosphere, the TOFCs were collected from the middle of the stack using cleaned, stainless-steel tweezers. The collected TOFC samples were immediately and individually wrapped with aluminum foil and stored at −20 °C until further analysis.

### 2.2. SEM Imaging

The morphology (shape) of the inner surface of the TOFCs was analyzed using a scanning electron microscope (SEM; Sigma 500; ZEISS, Berlin, Germany). The operating parameters of the SEM imaging were set as follows: accelerating voltage 5–15 kV, emission current 11,500 nA, probe current 50 mA, and working distance 9–12 mm. SEM images were obtained under different magnifications (50–350×) in different areas of each TOFC sample (Appendix A, Appendix A). 

### 2.3. Sample Treatment

The MPs were extracted from the TOFCs following previously reported methods [33,34], with some modification, however. The TOFC samples were firstly treated with flushing the inner wall of the TOFCs. In brief, 100 mL of pure water (at 25 °C) was transferred to the TOFCs, which were covered with the lids and then shaken at 60 rpm for 20 min. This procedure was repeated twice with the same method. After that, the inner wall of the TOFCs was secondly flushed with hot water. Briefly, 100 mL of pure water (at 95 °C) was added to the TOFCs, which had been flushed with pure water at 25 °C. These TOFCs were then covered with the lids and shaken at 60 rpm for 30 min. This procedure was also repeated twice. After that, all of the flushed pure water obtained in the two extraction procedures was combined (in total around 600 mL) and then filtrated through 50 μm glass fiber membranes (Merck-Millipore; Boston, MA, USA). After filtration, these filters were individually wrapped with aluminum foil and then dried in an oven at 60 °C. Notably, the influence of pH values of the food on the MP abundance in TOFCs was not considered, which is a limitation of this study [40].

### 2.4. Detection and Analysis of MPs

The MPs retained on the filters were picked out using a needle and tweezers and then examined using a stereomicroscope (SteREO Discovery. V8; Carl-Zeiss, Gottingen, Germany) in order to determine the physical characteristics of the MPs, including their shape and color [25]. Image J software was applied to measure the size of the MPs. All of the MPs retained on the filters were used to determine their chemical compositions by FT-IR spectrometry (Nicolet™ iS 50; Thermo-Scientific, MA, USA) operated under the attenuated total reflection mode. The IR wavelength range was set at 650–4500 cm^−1^ with the spectral resolution of 5 cm^−1^ for all samples. The obtained spectrums of all the particles were matched with the library spectrums provided by the OMNIC Picta software, and the particles with their spectrums having the matching degrees of >70% were identified as MPs. The TOFC sample processing diagram is shown in the Appendix A, Appendix A. The typical FT-IR spectra of the polymers of the MPs detected in the TOFCs are shown in the Appendix A, Appendix A. The detection limit of the instrument was 50 μm for the MPs.

### 2.5. Daily Intake Calculation for MPs

The estimated daily intake (EDI; items/kg bw/day) of MPs through using TOFCs was calculated for the general population on the basis of the following equation:$$\mathrm{EDI}=\frac{C_{\mathrm{TOFC}}\times UF\times N\times f}{W}$$
where *C*_TOFC_ represents the amount of MPs in TOFCs (items/ind.); *UF* means the use frequency of WFTOCs, and is set at 0.5 times/day (on average) for the general Chinese population; *N* is the number of TOFCs used, and it is assumed that at each time, on average, 2.5 TOFCs were used to wrap up food; *W* means the body weight of general adults, and was set at 55 and 65 kg for Chinese women and men, respectively [41]; *f* means the proportion of MPs in TOFCs that were ingested by humans every time, and is set at 25% based on the study of Du et al. (2020).

### 2.6. QA/QC 

Special precautions were taken to reduce the possibility of contamination of the MPs used throughout the whole experiment. No plastic consumables were used during the whole sample collection and extraction procedures. All glass containers were rinsed three times with pure water and completely dried at 80 °C before use [1]. All of the glass fiber filters were rinsed with pure water and then heated at 400 °C for 12 h prior to use [42]. Cotton lab coats and gloves were worn, and all solutions were always covered with aluminum foil throughout the whole experiment [43]. The pure water used for sample extraction was filtered with a 0.45 μm glass fiber membrane (GF/B Whatman, Maidstone, UK) before use. The collected TOFCs were covered with their own lids to prevent any MP contamination from the ambient atmosphere. In the TOFC sample treatment process, control experiments (pure water, *n* = 15) were also conducted to monitor the potential procedural contamination of the MPs. In this study, MPs were barely detected in the control experiments (*n* = 10), with an abundance of <2 items/sample. Despite this, any MPs detected in the control experiments were still deducted from the quantified MP abundance values for the TOFC samples. To validate the method of analysis used for the MPs, mixed MP standards (mix MPs ZK ldir; WEIPU Co., Shanghai, China) were analyzed using the current method. This standard included polyethylene (PE), polypropylene (PP), PS, polycarbonate, polyethylene terephthalate, polyurethane, and polyvinyl chloride (PVC). Results showed that all types of MPs could be identified, with a matching degree of >88%, which was not used for the quantification.

## 3. Results

### 3.1. Abundance of MPs in TOFCs

MPs were detected in all TOFC samples (*n* = 210) collected from seven Chinese cities, with an abundance ranging from 3 to 43 items/TOFC (Figure 1 and Table 1). The detailed MP concentrations in the TOFCs from the different Chinese cities are provided in the Table 1. The TOFCs from Chengdu (25 items/TOFC) contained the highest mean abundance of MPs, followed by Xining (23 items/TOFC), Dalian (16 items/TOFC), and Qingdao (14 items/TOFC). A relatively lower mean abundance of MPs was detected in the TOFCs from Hangzhou (11 items/TOFC), and Shanghai (8.7 items/TOFC). The MP concentrations in the TOFC samples were always non-normally distributed (Shapiro–Wilk test, *p* < 0.05). The coefficient of mean variation (calculated as standard deviation divided by mean) for the MP abundance in the TOFCs from the different cities was in the range of 36 (Chengdu)–62% (Qingdao). 

### 3.2. Shape and Size of MPs in TOFCs

Fiber was consistently the major shape of the MPs in the TOFC samples, accounting for 66–87% of the total shapes of the MPs in the TOFCs from the different cities (Figure 2 and Appendix A, Appendix A). The highest mean proportion of fiber was observed in the TOFCs taken from Qingdao (87%), followed by those from Xining (83%), Guangzhou (76%), and Chengdu (75%). Fragment (mean proportion 14–28%) was the second major shape of the MPs in the TOFC samples, except for the MPs from the Qingdao TOFC samples, in which film (6.5%) had a comparable mean proportion to fragment (6.2%).

The size distribution of MPs in TOFCs displayed no obvious differences among different cities (Figure 2 and Appendix A, Appendix A). The recorded minimum and maximum size of MPs were 50 and 5000 μm, respectively. Most of the MPs in the TOFCs from the different cities had a size of 201–500 μm, except those from Qingdao, and the MPs having a size of 201–500 μm accounted for a mean 34–42% of the total MPs in the TOFCs. The major size of the MPs in the TOFCs from Qingdao was in the range of 101–200 μm. In addition, 501–1000 μm was the second major size of the MPs in the TOFCs from Chengdu, Dalian, and Guangzhou, accounting for a mean 23, 25, and 23% of the total MPs, respectively. Furthermore, 101–200 μm was the second major size of the MPs in the TOFCs from the remaining cities, contributing a mean 22–30% of the total MPs.

### 3.3. Color and Polymer Composition of MPs in TOFCs

Consistently, transparent was the major color type of the MPs in the TOFCs from the different Chinese cities, with the mean proportion of 39 (Qingdao)–73% (Hangzhou) of the total MPs (Figure 3 and Appendix A, Appendix A). In general, the colors of the MPs in the TOFCs from Qingdao, Chengdu, Xining, and Dalian were comparatively more diverse than those from Guangzhou, Shanghai, and Hangzhou. Specifically, transparent (mean 39–56%), white (12–29%), black (8.3–22%), and blue (9.3–17%) were the major colors of the MPs in the TOFCs from Qingdao, Chengdu, Xining, and Dalian. Nevertheless, for the MPs in the TOFCs from Guangzhou, Shanghai, and Hangzhou, transparent and white were consistently the predominant color types, and they collectively accounted for a mean 86, 93, and 95% of the total MPs, respectively.

The percentage of identified MPs to all suspected particles in the TOFC samples was in the range of 30–68%. More than ten polymer compositions were identified for the MPs in the TOFCs (Figure 4 and Appendix A, Appendix A), and the most commonly detected polymers were polypropylene (PP), polyethylene (PE), and polyester. Polymer compositions of the MPs in the TOFCs from the different cities were consistently dominated by PP, which represents a mean 56 (Dalian)–73% (Hangzhou) of the total MPs. Polyethylene was the second predominant polymer composition of the MPs in the TOFCs from Guangzhou (a mean 15%), Hangzhou (19%), and Shanghai (20%). However, polyester was the second major polymer composition of the MPs in those TOFCs from Chengdu (a mean 9.4%) and Dalian (17%). Other polymer compositions of the MPs in total contributed <20% of the total MPs in the TOFC samples.

### 3.4. Estimated Daily Intake of MPs

Based on the abundance of MPs in the TOFCs, we estimated the daily intake of MPs through using TOFCs for the general Chinese population (Table 2). Among the seven Chinese cities, the highest mean EDI of MPs was calculated for Chengdu (0.14 and 0.12 items/kg bw/day for women and men, respectively), which is similar to that for Xining (0.13 and 0.11 items/kg bw/day), followed by Dalian (0.088 and 0.075 items/kg bw/day) and Qingdao (0.080 and 0.067 items/kg bw/day). The lowest mean EDI was observed for Shanghai (0.049 and 0.042 items/kg bw/day), which is approximately three times lower than that for Chengdu. Moreover, women consistently had higher EDIs of MPs than men. 

## 4. Discussion

Limited studies have carefully investigated the presence of MPs in plastic takeout food containers that are still being widely used in China. In this study, we characterized the pollution of MPs in TOFCs collected from seven Chinese cities. The abundance of the MPs in the TOFCs showed a great difference among the different cities, with the TOFCs from Chengdu (mean 25 items/TOFC) having a mean MP abundance approximately three times higher than those from Shanghai (8.7 items/TOFC). Du et al. (2020) had reported an abundance of mean 9 items/container for MPs in PP food containers from China, which is comparable to the level observed in the TOFCs from Shanghai. Microplastics in all seven kinds of takeout food containers collected from supermarkets in Hangzhou had an abundance of 29–552 items/container [44], which is much higher than that reported in this study. Takeout food containers are manufactured through the pressurized injection of melted PP raw materials to the molding bed, followed by natural cooling. Different manufacturing processes may result in different characteristics of the inner walls of TOFCs, which may contribute to the different levels of abundance of MPs released from the TOFCs. In contrast, the finding of Du et al. (2020) showed that differences among MP abundance in plastic food containers from five Chinese cities were not significant. This is in contrast with our study, which is partially owing to the differences in study design and sampling campaign. 

Fiber was consistently the predominant shape of the MPs in the TOFCs from the different Chinese cities. Despite fragment-shaped MPs only comprising 6.2–28% of the total MPs on average, fragment was still the second major shape of the MPs in the TOFCs. The fiber MPs in the TOFCs may be mainly from the deposition of atmospheric MPs during the production, storage, and transportation of TOFCs, since fiber has always been the major shape of MPs in indoor atmosphere and dust from Chinese cities [45,46]. Most of the MPs in the TOFCs collected from the different Chinese cities had the size of 201–500 μm. These shape and size profiles of the MPs are similar to the results of Du et al. (2020), which also found that fiber MPs accounted for >50% of the total MPs in plastic food containers, with the major size of 30–500 μm. In contrast, monitoring studies have shown an average diameter of 50 nm for the plastic debris in PP-made food containers from Beijing, China, with the shape of MPs dominated by fragment [33]. In that study, SEM was employed to analyze the morphology of MPs retained on the inner surface of containers. We speculate that distinctions in the methods of analysis may contribute to the observed discrepancy.

The colors of the MPs were relatively more diverse in the TOFCs from Qingdao, Chengdu, Xining, and Dalian; transparent was always the major color type of the MPs in the TOFCs. The microplastics in the TOFCs consisted of a variety of polymers, with PP being the dominant polymer composition of the TOFCs. Colors and chemical compositions of MPs are similar to that of the synthetic materials of TOFCs. This suggests that most of the MPs were derived from the inner surface of containers. Alternatively, the prolonged photobleaching of colored MPs may also result in transparent MPs being observed in TOFCs [42]. Different storage times and conditions may result in different colors of MPs being detected in TOFCs.

Either PE or polyester was the second major polymer composition of the MPs in the TOFCs. Polyethylene is one of the most commonly used polymers worldwide, and is widely used to produce consumer products, such as plastic bags and food containers [47,48]. Polyester fiber has been the raw material for producing various cloths, textiles, and carpets [49]. These fibers are largely leached out during their use, and washing and wearing processes as PE and polyester MPs, and eventually enter the environment [50,51]. Polyester and PP MPs have been widely detected in indoor atmosphere and dust from China [25,46,47]. We speculate that the PP and polyester the MPs in the TOFCs were likely from the atmospheric microplastic pollution, which was generated during the manufacture, storage, and transport of these TOFCs. This may be significantly different between the various cities in China. In addition, manufacturers may produce TOFCs using recycled PP materials that may contain residual PE. This may lead to the occurrence of PE MPs on the inner surface of TOFCs. Overall, differences in manufacturing methods, storage conditions, and transport ways of TOFCs may lead to different polymer composition of MPs in TOFCs.

The takeout food-delivery business is growing rapidly in China, and Chinese takeout-food customers had reached 358 million people in 2018 [52]. Most of these customers are white-collar workers in offices, with an average ordering frequency of 4–7 times/week [34] In this study, the human intake of MPs after using TOFCs was calculated for the general Chinese population based on the abundance of the MPs in the TOFC samples and the use frequency of TOFCs. Women had comparatively higher EDIs of MPs than men, which is mainly due to the lower body weight of women. Many factors could influence the amount of daily MP intake for humans, such as eating habits and lifestyles. Notably, these factors were not taken into consideration in the current EDI calculation, and this is a limitation of this study. Instrument limits (50 μm for MPs) may result in the underestimation of the human daily intake of MPs.

Indoor dust inhalation has been deemed as a significant source of human exposure to MPs [48]. Comparatively, the calculated EDI of MPs through using TOFCs is lower than that reported for general Chinese adults through indoor dust inhalation of MPs (mean 0.23 items/kg bw/day, range 0.020–0.56 items/kg bw/day) [25]. Liu et al. (2019) reported concentrations of suspended atmospheric MPs in Shanghai, China, and estimated that the intake of MPs was 21 items/day for the general population, which is higher than that reported in this study (corresponding to 2.7–7.8 items/day). The amount of human exposure to MPs through other sources has also been estimated. For example, through inhalation, table salt, and tap water ingestion, humans were expected to intake 25,575, 3000, and 2784 items of MPs, respectively, every year [34,53,54]. Accordingly, 986–2847 items of MPs were ingested through using TOFCs each year, which is comparable to the intake through drinking tap water.

## 5. Conclusions

This study investigated the occurrence of MPs in TOFCs from China. Microplastics were detected in all TOFC samples collected from seven Chinese cities, with an abundance of 3–43 items/TOFC. The TOFCs from Chengdu contained the highest mean abundance of MPs, followed by Xining, Dalian, Qingdao, Guangzhou, Hangzhou, and Shanghai. Fiber was consistently the major shape of the MPs in the TOFC samples. Most of the MPs in the TOFCs from the different cities had a size of 201–500 μm, except those from Qingdao. Consistently, transparent was the major color type of the MPs in the TOFCs from the different cities, and the colors of the MPs in the TOFCs from Qingdao, Chengdu, Xining, and Dalian were comparatively more diverse than those from Guangzhou, Shanghai, and Hangzhou. More than ten polymer compositions were identified for the MPs in the TOFCs, and the polymer composition of the MPs in the TOFCs from the different cities was consistently dominated by PP. The mean EDI of MPs through using TOFCs for the general Chinese population was in the range of 0.042–0.14 items/kg bw/day. In addition, more studies are still warranted to explore the presence of nanoparticles in TOFCs from China.

## Figures and Tables

**Figure 1 toxics-10-00762-f001:**
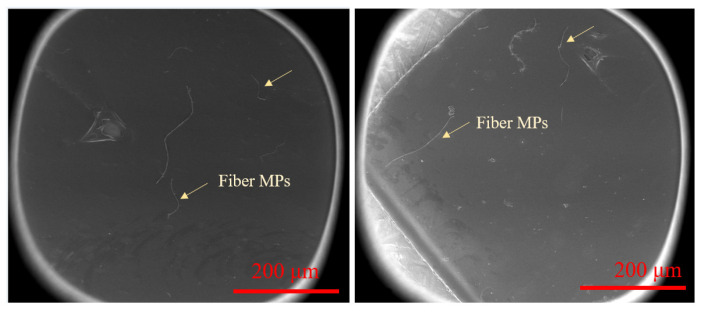
SEM images of fiber MPs on the inner surface of TOFCs.

**Figure 2 toxics-10-00762-f002:**
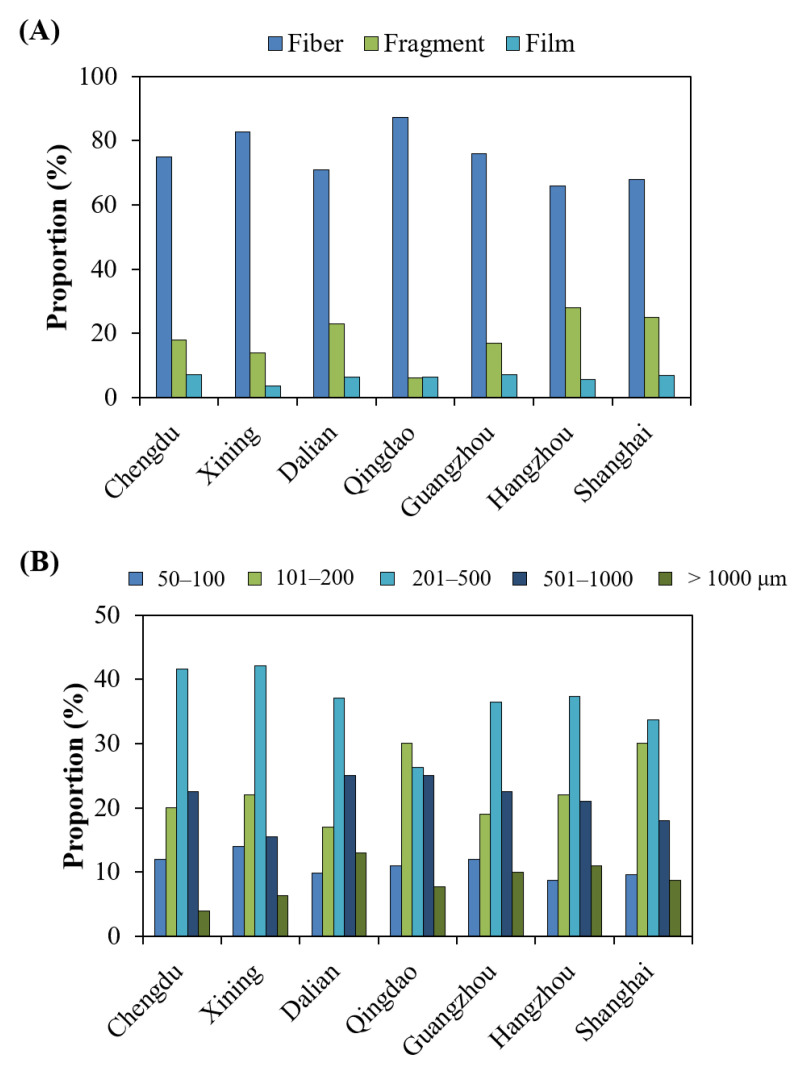
Percentage (%) of different (**A**) shapes and (**B**) sizes of MPs in takeout food containers collected from different Chinese cities.

**Figure 3 toxics-10-00762-f003:**
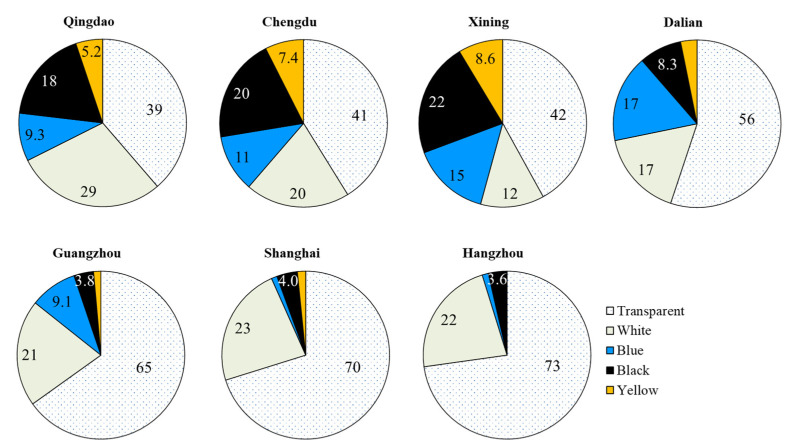
Percentage (%) of different colors of MPs in takeout food containers collected from different Chinese cities.

**Figure 4 toxics-10-00762-f004:**
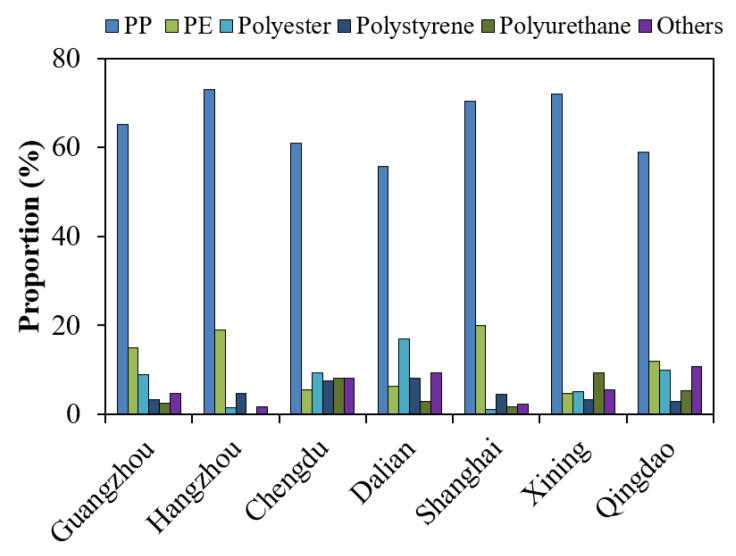
Polymer composition of MPs in takeout food containers collected from different cities in China. “Others” means other polymer compositions of MPs, mainly including polyethylene chlorinated, polyamide, polycarbonate, and polyethylene terephthalate. PP and PE indicate polypropylene and polyethylene, respectively.

**Table 1 toxics-10-00762-t001:** Abundance (Items/TOFC) of MPs in TOFCs collected from Chinese cities.

Sampling Site	Mean	Median	SD	Range
Chengdu (*n* = 27)	25	27	9.3	10–43
Xining (*n* = 37)	23	26	10	9–40
Dalian (*n* = 32)	16	12	7.5	7–27
Qingdao (*n* = 28)	14	18	8.7	7–21
Guangzhou (*n* = 21)	13	11	6.0	6–28
Hangzhou (*n* = 35)	11	9.7	5.5	8–16
Shanghai (*n* = 30)	8.7	10	4.4	3–18

**Table 2 toxics-10-00762-t002:** Estimated daily intake of MPs (items/kg bw/day) through using TOFCs for general Chinese population in different Chinese cities.

	Chengdu	Xining	Dalian	Qingdao	Guangzhou	Hangzhou	Shanghai
	*women*						
Mean	0.14	0.13	0.088	0.080	0.071	0.063	0.049
Median	0.16	0.15	0.091	0.077	0.082	0.058	0.050
Range	0.057–0.24	0.051–0.23	0.040–0.15	0.040–0.12	0.034–0.16	0.045–0.091	0.017–0.10
	*men*						
Mean	0.12	0.11	0.075	0.067	0.060	0.053	0.042
Median	0.13	0.12	0.080	0.072	0.058	0.054	0.046
Range	0.048–0.21	0.043–0.19	0.034–0.13	0.034–0.10	0.029–0.13	0.038–0.077	0.014–0.087

## Data Availability

Not applicable.

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
