# Peer review of "Microplastics in Widely Used Polypropylene-Made Food Containers"

_toxics, 2022, doi:10.3390/toxics10120762_

Round 1
Reviewer 1 Report
Thank you for the opportunity to review this study.
This manuscript reveals new results to the scientific community in MP analysis and I appreciate the scientific work on MPs in polypropylene-made takeout food containers. However, some parts need to be modified, especially in the introduction and materials and methods paragraphs. The novelty of this work is not well described, and it is confusing. In the introduction part, it is necessary to explain better the limitation of this topic in the scientific literature, in order to understand the aim of this study. Also, in the materials and methods part, more details on the technique and the methods employed for the quantification and identification are missing. Thus, more essential concepts are missing, the authors may clarify significantly all the paragraphs and I highly recommend a major revision for this manuscript before acceptance. If the authors would not clarify the issues mentioned above, I will not recommend that this study has enough quality to warrant publication in its existing form.
Introduction
-“ which are defined as plastic particles with a diameter of < 5.0 mm (Jiang et al. 2020; Birch et al. 2020)”. This definition is too generic, and it is not clearly explained. Please add more recent definitions in terms of size (defined also the smallest limit of the definition of MPs) and typology, for instance from ECHA.
- “Poly-propylene plastic is manufactured by polymerizing the propylene monomer”, please use acronyms (PP) as the authors said before.
-The authors said that “Moreover, several studies have isolated and characterized MPs in PP-made food containers from China” and after ten lines “few studies have characterized the occurrence of MPs in various plastic food containers collected from China”. It is not clear if the knowledge of MPs from food containers is limited and /or unknown.
- “ the spectroscopic technique”, the authors may be inserted the typology of the technique (FTIR, Raman?).
-“However, these previous studies usually included a relatively small sample size,” it could be helpful in understanding the range of the number of sample that they are considering comparing with the scientific literature.
- from “However, these previous studies usually included a relatively small sample size, to…. exposure to MPs, and for risk assessment on human exposure to MPs”. This part is not clear, and the authors may clarify it better to consequently also understand the aim of their study.
Hence, as I read in the introduction, other studies have just analyzed MPs from TOFC; however, the authors said that these limited studies “did not analyze the characteristics (e.g., size, color, shape, and chemical composition)”. I’m reading the study from He et al., 2021 and they used Raman spectroscopy and fluorescence microscopy for qualitative analysis. Also, I read from Du et al., 2020, that physical characteristics including shapes and colors were recorded. Moreover, “chemical compositions of the particles were identified by μ-FTIR (Thermo Scientific, Nicolet iN10, USA) using the transmission mode”. It was the same in the study of Fadare et al., 2020, where they used FTIR and SEM for MPs analysis from food containers.
Hence, the introduction part needs to be rewritten completely and the authors should explain better the novelty of their work compared with the existing studies and their limitations.
Materials and Methods
- Why did the authors explain the instrument analysis in the Sample collection part?
- For SEM analysis: What do the authors mean by “Morphology analysis ”? They may describe if they evaluated the size or shape…
- In Sample treatment: The authors may use “however”, instead of “but”.
- Detection and analysis:
The authors said, “All of the MPs retained on the filters were used to determine their chemical compositions by FT-IR spectrometry (Nicolet™ iS 50; Thermo-Scientific, MA, USA), operated under the attenuated total reflection mode”. Did the authors transfer ALL MPs counted with a stereomicroscope in the ATR support? How? Also, the smallest size (50-100 µm)? It is quite difficult to transfer these such of the smallest particles. Did the authors count all MPs on the surface of the filter? A detailed explanation of the quantification and chemical identification of MPs is missing.
Please provide the minimum size in length and width that the authors achieved with this technique. Please add also the “libraries” that were used with Omnic software (in Supplementary Materials).
- From Fig. S4 “Typical FT-IR spectra of the polymers of MPs detected in TOFCs”. It could be more useful to add in the same picture the spectra obtained from MPs in TOFCs with the comparison of the respective spectra from the library (including the match percentage).
-“ Rayon and cellophane were deemed as MPs in this study”. Why? Rayon is a semi-synthetic fiber, made from natural sources of regenerated cellulose, while cellophane is made of regenerated cellulose. I do not agree with this decision. The authors may be included more references and explanations why they decided to incorporate these substances in the MPs group. I suggest splitting the two categories separately.
- regarding the standards: “Results showed that all types of MPs could be identified, with the matching degree of > 88%”. Did the authors verify only the identification matches from the use of standard? Did they use them also for the quantification (recovery rate as the number of MPs)? It is not cleared.
-Regarding QA/QC: “In the TOFC sample treatment process, control experiments were also conducted to monitor the potential procedural contamination of MPs”. How many controls? did they use pure water? Explain better.
Results:
-From the results, I read that “Detection limitation of the instrument (50 μm for MPs)”. Is it the limitation of the filter porosity used or the limit of the instrument? It is very important to explain better un Materials and Methods the limitation of the MPs size in this study.
-From table 1, I can see that the SD of each sample is quite high. Did the authors try to use other error measures, as the data did not follow a normal distribution?
Conclusions
- “This is the most comprehensive study, to our knowledge, investigating the occurrence of MPs in TOFCs from China”. What did the authors mean by “comprehensive”? The number of Chinese cities? I did not find significant novelty compared to the other similar studies that were cited in the introduction.

Author Response
Thank you for the opportunity to review this study.
This manuscript reveals new results to the scientific community in MP analysis and I appreciate the scientific work on MPs in polypropylene-made takeout food containers. However, some parts need to be modified, especially in the introduction and materials and methods paragraphs. The novelty of this work is not well described, and it is confusing. In the introduction part, it is necessary to explain better the limitation of this topic in the scientific literature, in order to understand the aim of this study. Also, in the materials and methods part, more details on the technique and the methods employed for the quantification and identification are missing. Thus, more essential concepts are missing, the authors may clarify significantly all the paragraphs and I highly recommend a major revision for this manuscript before acceptance. If the authors would not clarify the issues mentioned above, I will not recommend that this study has enough quality to warrant publication in its existing form.
Response: we thank the reviewer very much for your comments. We have revised the manuscript based on your comments.
Introduction
-“ which are defined as plastic particles with a diameter of < 5.0 mm (Jiang et al. 2020; Birch et al. 2020)”. This definition is too generic, and it is not clearly explained. Please add more recent definitions in terms of size (defined also the smallest limit of the definition of MPs) and typology, for instance from ECHA.
Response: as suggested, we have revised the definition of MPs: “which are defined as plastic particles with the diameter of 1 nm–5 mm (length of 3 nm–15 mm and length to diameter ratio of > 3 for fibers)”.
- “Poly-propylene plastic is manufactured by polymerizing the propylene monomer”, please use acronyms (PP) as the authors said before.
Response: we have revised that, thanks.
-The authors said that “Moreover, several studies have isolated and characterized MPs in PP-made food containers from China” and after ten lines “few studies have characterized the occurrence of MPs in various plastic food containers collected from China”. It is not clear if the knowledge of MPs from food containers is limited and /or unknown.
Response: the knowledge of MPs from food containers is limited. We have revised that as “limited studies have isolated and characterized MPs in PP-made food containers from China”.
- “ the spectroscopic technique”, the authors may be inserted the typology of the technique (FTIR, Raman?).
Response: we have added “FTIR” in the manuscript.
-“However, these previous studies usually included a relatively small sample size,” it could be helpful in understanding the range of the number of sample that they are considering comparing with the scientific literature.
Response: as suggested, we have added the number of samples in the manuscript: “In this study, polypropylene-made TOFCs (n = 210) used for fast food packaging were collected from restaurants in seven Chinese cities”.
- from “However, these previous studies usually included a relatively small sample size, to…. exposure to MPs, and for risk assessment on human exposure to MPs”. This part is not clear, and the authors may clarify it better to consequently also understand the aim of their study.
Response: as suggested, we have added the aim of this study in the manuscript: “with the aim to investigate the abundance, shape, size, color, and chemical composition of MPs in TOFCs. Human daily intake of MPs was also estimated for the general Chinese population, based on the abundance of MPs in TOFCs.”
Hence, as I read in the introduction, other studies have just analyzed MPs from TOFC; however, the authors said that these limited studies “did not analyze the characteristics (e.g., size, color, shape, and chemical composition)”. I’m reading the study from He et al., 2021 and they used Raman spectroscopy and fluorescence microscopy for qualitative analysis. Also, I read from Du et al., 2020, that physical characteristics including shapes and colors were recorded. Moreover, “chemical compositions of the particles were identified by μ-FTIR (Thermo Scientific, Nicolet iN10, USA) using the transmission mode”. It was the same in the study of Fadare et al., 2020, where they used FTIR and SEM for MPs analysis from food containers.
Hence, the introduction part needs to be rewritten completely and the authors should explain better the novelty of their work compared with the existing studies and their limitations.
Response: thanks for the suggestion. We have rewritten and added some content in the introduction part: “To our knowledge, few studies have characterized the oc-currence of MPs in various plastic food containers collected from China (Du et al. 2020; Fadare et al. 2020; He et al. 2021). For in-stance, employing the spectroscopic technique (FT-IR), Fadare et al. (2020) reported that the average weight of isolated MPs was 3−38 mg/pack in consumer PP-made food containers (n = 150) from Beijing, China. Du et al. (2020) analyzed takeout food con-tainers (n = 60) made of 4 common polymers from five Chinese cities, and reported that the abundance of MPs ranged from 3 to 29 items/container. However, these previous studies usually in-cluded a relatively small sample size, and did not analyze the characteristics (e.g., size, color, shape, and chemical composition) of MPs. More studies with a larger sample size are still war-ranted to comprehensively understand the pollution of MPs in PP-made TOFCs from China. Such information is necessary for understanding the sources of human oral exposure to MPs, and for risk assessment on human exposure to MPs.”
Materials and Methods
- Why did the authors explain the instrument analysis in the Sample collection part?
Response: this is because the instrument analysis method is used to check the chemical compositions of TOFCs.
- For SEM analysis: What do the authors mean by “Morphology analysis ”? They may describe if they evaluated the size or shape…
Response: in fact, we evaluated the shape. So, we have revised that as: “Morphology (shape) of the inner surface of TOFC was analyzed using the scanning electron microscope (SEM; Sigma 500; ZEISS, Berlin, Germany).”
- In Sample treatment: The authors may use “however”, instead of “but”.
Response: we have replaced “but” with “however”.
- Detection and analysis:
The authors said, “All of the MPs retained on the filters were used to determine their chemical compositions by FT-IR spectrometry (Nicolet™ iS 50; Thermo-Scientific, MA, USA), operated under the attenuated total reflection mode”. Did the authors transfer ALL MPs counted with a stereomicroscope in the ATR support? How? Also, the smallest size (50-100 µm)? It is quite difficult to transfer these such of the smallest particles. Did the authors count all MPs on the surface of the filter? A detailed explanation of the quantification and chemical identification of MPs is missing.
Response: as suggested, we have added such information in the manuscript: “MPs retained on the filters were picked out using a needle and tweezers, and then examined using a stereomicroscope”.
Please provide the minimum size in length and width that the authors achieved with this technique.
Please add also the “libraries” that were used with Omnic software (in Supplementary Materials).
Response: as suggested, we have added the minimum size in the manuscript: “Detection limit of the instrument was 50 μm (length) for MPs.”
- From Fig. S4 “Typical FT-IR spectra of the polymers of MPs detected in TOFCs”. It could be more useful to add in the same picture the spectra obtained from MPs in TOFCs with the comparison of the respective spectra from the library (including the match percentage).
Response: we have provided the spectra from the library (including the match percentage).
Figure S4. Typical FT-IR spectra and matching degree of the polymers of MPs detected in TOFCs.
-“ Rayon and cellophane were deemed as MPs in this study”. Why? Rayon is a semi-synthetic fiber, made from natural sources of regenerated cellulose, while cellophane is made of regenerated cellulose. I do not agree with this decision. The authors may be included more references and explanations why they decided to incorporate these substances in the MPs group. I suggest splitting the two categories separately.
Response: we agree with the reviewer. However, using our method, we can not accurately separate these two MPs. So, we did not split the two categories separately in this study.
- regarding the standards: “Results showed that all types of MPs could be identified, with the matching degree of > 88%”. Did the authors verify only the identification matches from the use of standard? Did they use them also for the quantification (recovery rate as the number of MPs)? It is not cleared.
Response: we did not use them for the quantification: “Results showed that all types of MPs could be identified, with the matching degree of > 88%, which was not used for the quantification.”
-Regarding QA/QC: “In the TOFC sample treatment process, control experiments were also conducted to monitor the potential procedural contamination of MPs”. How many controls? did they use pure water? Explain better.
Response: we have added that information: “In the TOFC sample treatment process, control experiments (pure water, n = 15) were also conducted to monitor the potential procedural contamination of MPs.”.
Results:
-From the results, I read that “Detection limitation of the instrument (50 μm for MPs)”. Is it the limitation of the filter porosity used or the limit of the instrument? It is very important to explain better un Materials and Methods the limitation of the MPs size in this study.
Response: we have revised it as: “Limit of the instrument (50 μm for MPs) may result in the underestimate of human daily intake of MPs.”
-From table 1, I can see that the SD of each sample is quite high. Did the authors try to use other error measures, as the data did not follow a normal distribution?
Response: we have tried using relative standard deviation, but which is still quite large.
Conclusions
- “This is the most comprehensive study, to our knowledge, investigating the occurrence of MPs in TOFCs from China”. What did the authors mean by “comprehensive”? The number of Chinese cities? I did not find significant novelty compared to the other similar studies that were cited in the introduction.
Response: thanks for the suggestion. We have revised that as: “This study investigated the occurrence of MPs in TOFCs from China.”

Reviewer 2 Report
Review of “Microplastics in Widely Used Polypropylene-Made Food Containers”
This is a worthy study and we need to understand the release load of microplastics from food containers to assess the health effects. However, there are major problems with this study. It focuses too much on the relationship of different cities and the amount of microplastics from the TOFCs instead of relating the data to the specific types of containers. THis is not helpful: “Based on the identification code information of the TOFCs, it is believed that the majority of TOFCs are made in different ways.” The properties of the containers are important for understanding the release of MP in food. Do different cities utilize different plastic food containers? Further, the authors speculate on relationships that can be measured and do not utilize control experiments to better understand the sources of the synthetic microfibers. Are fibers formed from the forceful shaking? Or are fibers contaminating the containers in the production and distribution stages?
There are numerous grammar errors that reduces the quality of the manuscript.
What is the importance of selecting coastal and inland cities? Do these populations use takeout containers to a different extent?
In the Sample Collection section, why do the authors discuss methods for plastic container identification? Why is there an italicized section on SEM imaging in this section?
How is this known? “Production date of TOFCs was between July 2019 and June 2020.”
What was the size of the containers? What specifically are the function of the containers? For hot food, cold food? Were they all the same shape, thickness, color, etc?
“Notably, the influence of pH values of the food on the MP abundance in TOFCs was not considered, which is a limitation of this study.” Was the food packaged in the container water-based or oil/fat-based? Recent published work suggests this is a significant factor: https://www.sciencedirect.com/science/article/pii/S0269749122013859 .
Can the authors provide a citation for this information? “0.5 times/day (on average) for the general Chinese population”
PMFCs?
Can the authors clarify this? “In the TOFC sample treatment process, control experiments were also conducted to monitor the potential procedural contamination of MPs.”
The authors should clearly explain that the methodology flushes the plastic containers with water with force to determine the “MPs detected in all TOFC samples.” This is a clean and simple way to assess and estimate the release of microplastics from the food containers, but it is does not consider contents and other factors. Have any studies shown that this method releases an amount/type of microplastics that is comparable to their actual use?
Again, please explain why the data were grouped and analyzed this way.
“Consistently, transparent was the major color of MPs in TOFCs” Were all the analyzed containers transparent?
Be more specific than “polyester”
Observed or calculated from the data? “the highest mean EDI of MPs was observed for Chengdu”
Did the cited researchers use the same methodology? “MPs in all 7 kinds of takeaway food containers collected from supermarkets in Hangzhou had the abundance of 29–552 items/container (Zhou, et al. 2022), which is much higher than that reported in this study.”
Do the authors mean PE? “We speculate that PP and polyester MPs in TOFCs were likely from the atmospheric microplastic pollution,..”
Detection limits should be stated in the methods section. “Detection limitation of the instrument (50 μm for MPs) may result in the underestimate of human daily intake of MPs.”
Author Response
This is a worthy study and we need to understand the release load of microplastics from food containers to assess the health effects. However, there are major problems with this study. It focuses too much on the relationship of different cities and the amount of microplastics from the TOFCs instead of relating the data to the specific types of containers. THis is not helpful: “Based on the identification code information of the TOFCs, it is believed that the majority of TOFCs are made in different ways.” The properties of the containers are important for understanding the release of MP in food. Do different cities utilize different plastic food containers? Further, the authors speculate on relationships that can be measured and do not utilize control experiments to better understand the sources of the synthetic microfibers. Are fibers formed from the forceful shaking? Or are fibers contaminating the containers in the production and distribution stages?
Response: we thank the reviewer vey much for your comments.
There are numerous grammar errors that reduces the quality of the manuscript.
Response: the manuscript has been revised by a native English speaker.
What is the importance of selecting coastal and inland cities? Do these populations use takeout containers to a different extent?
Response: yes. In China, populations in coastal cities use more takeout containers.
In the Sample Collection section, why do the authors discuss methods for plastic container identification? Why is there an italicized section on SEM imaging in this section?
Response: we want to identify the polymer composition of plastic containers, so we discussed methods for plastic container identification. Right now, SEM Imaging section has been seperated from the Sample Collection section.
How is this known? “Production date of TOFCs was between July 2019 and June 2020.”
Response: that is a mistake, and we have deleted that statement.
What was the size of the containers? What specifically are the function of the containers? For hot food, cold food? Were they all the same shape, thickness, color, etc?
Response: size of the containers has been added in the manuscript: “The mean size of TOFC was 27 cm × 11 cm × 5.7 cm”. TOFCs were mainly used to wrap up fast food, including cooked rice, meat, vegetables, noodles, and soups, in these restaurants. they were all in the same or simialr shape, thickness, color, etc.
“Notably, the influence of pH values of the food on the MP abundance in TOFCs was not considered, which is a limitation of this study.” Was the food packaged in the container water-based or oil/fat-based? Recent published work suggests this is a significant factor: https://www.sciencedirect.com/science/article/pii/S0269749122013859 .
Response: thanks for the suggestion. We have cited this published work in the manuscript.
Can the authors provide a citation for this information? “0.5 times/day (on average) for the general Chinese population”
Response: this is our estimation.
PMFCs?
Response: that is a mistake, and we have revised it as “TOFCs”
Can the authors clarify this? “In the TOFC sample treatment process, control experiments were also conducted to monitor the potential procedural contamination of MPs.”
Response: we have added that information: “In the TOFC sample treatment process, control experiments (pure water, n = 15) were also conducted to monitor the potential procedural contamination of MPs.”.
The authors should clearly explain that the methodology flushes the plastic containers with water with force to determine the “MPs detected in all TOFC samples.” This is a clean and simple way to assess and estimate the release of microplastics from the food containers, but it is does not consider contents and other factors. Have any studies shown that this method releases an amount/type of microplastics that is comparable to their actual use?
Response: our method was adapted from previous studies, which is suitable to estimate the release of microplastics from the food containers. We have added this in the manuscript: “MPs were extracted from TOFCs, following previously re-ported methods (Fadare et al. 2020; Du et al. 2020),”
Again, please explain why the data were grouped and analyzed this way.
Response: data were grouped and analyzed based on studies of Fadare et al. 2020 and Du et al. 2020
“Consistently, transparent was the major color of MPs in TOFCs” Were all the analyzed containers transparent?
Response: no. The analyzed containers were white.
Be more specific than “polyester”
Response: we have revised it as “polyester polymer”.
Observed or calculated from the data? “the highest mean EDI of MPs was observed for Chengdu”
Response: it was calculated from the data. So, we have revised it as: “the highest mean EDI of MPs was calculated for Chengdu”.
Did the cited researchers use the same methodology? “MPs in all 7 kinds of takeaway food containers collected from supermarkets in Hangzhou had the abundance of 29–552 items/container (Zhou, et al. 2022), which is much higher than that reported in this study.”
Response: yes. They used the same methodology.
Do the authors mean PE? “We speculate that PP and polyester MPs in TOFCs were likely from the atmospheric microplastic pollution,..”
Response: yes. We have revised that.
Detection limits should be stated in the methods section. “Detection limitation of the instrument (50 μm for MPs) may result in the underestimate of human daily intake of MPs.”
Response: as suggested, we have moved that in the methods section: “Detection limit of the instrument was 50 μm for MPs.”.

Author Response
Dear Editor,
I have read the article entitled Microplastics in Widely Used Polypropylene-Made Food containers. Here are my comments.
The topic is interesting, the article is well written and it is fluently readable. If I understand correctly, the article deals with a current state of microplastic content in containers after their manufacture. The question I have is how much sense then it makes to calculate the daily human intake and to compare the different Chinese cities with each other, as the situation could probably be different each time.
Response: we thank the reviewer very much for your comments.
The authors cite atmospheric input as the main MP source, which is certainly highly variable. I think that the authors should have put more emphasis on this currently situation, which may also change over time. As the authors note, the significant differences between the various cities in China, this fact should be better linked to the potential sources of these microplastics, the differences in the ways in which these products are produced, stored and generally handled.
Response: we agree that it is certainly highly variable. We have added some discussions in the manuscript: “We speculate that PP and polyester MPs in TOFCs were likely from the atmospheric microplastic pollution, which was generated during the manufacture, storage, and transport of these TOFCs. This may be significantly different between the various cities in China.”
In addition to this main comment, I have a few minor and more specific comments. It is interesting that in articles by Chinese authors on microplastics, cellophane is often mentioned as one of the plastic materials, as it is in the present article. This is largely absent in the articles by other authors. My question therefore is-how did the authors distinguish cellophane and rayon from cellulose, since both materials are modified cellulose, with practically identical IR spectra. Given that a large proportion of the MP particles consisted of filaments, and were transparent on top of that, it would have been good if the authors had clarified this a bit more in the paper. If we can already associate rayon with clothing, it is hard to do so with cellophane. And often the transparent filaments are of natural origin (cellulose).
Response: cellophane and rayon were not detected in this study, so we have deleted that statement.
As PP is the most abundant material-what was the shape of these particles. More could be written on this, especially linking the form and material of MP.
Response: as suggested, we have added some discussion in the manuscript: “Polyester and PP MPs have been widely detected in indoor atmosphere and dust from China (Zhu et al. 2022; Liao et al. 2021; Zhang, Zhao, et al. 2020). We speculate that PP and polyester MPs in TOFCs were likely from the atmospheric microplastic pollution, which was generated during the manufacture, storage, and transport of these TOFCs. This may be significantly different between the various cities in China. In addition, manufacturers may use recycled PP materials that may contain residual PE to produce TOFCs. This may lead to the occurrence of PE MPs on the inner surface of TOFCs. Overall, differences in manufacturing method, storage condition, and transport ways of TOFCs may lead to different polymer composition of MPs in TOFCs.”
Figure 1B and Table 1 show similar data, one is redundant.
Response: we have deleted Figure 1B.
I would advise that instead of male and female the authors use men and women.
Response: as suggested, we have replaced “male and female” with “men and women”.
Page 11, line 18: perhaps PE instead of PP?
Response: we have revised that.
Chlorinated PE is most often determined due to interference from the glass filter, especially when analysing filaments which are very thin and it is difficult to get a clean spectrum of the particle itself. Authors should clarify this.
Response: chlorinated PE was not detected in this study.
In conclusion, I think the paper brings interesting results, but some revision would be needed before publication.
Response: thanks for the suggestion again.

Round 2
Reviewer 1 Report
Thank you for the author's response. I now accept this manuscript in its present form.